# Prevalence and causes of ocular disorders and visual impairment among preterm children in Ethiopia

Sadik Taju Sherief  ,[1,2] Lulu M Muhe,[3] Amha Mekasha,[3] Asrat Demtse,[4] Asim Ali[5,6]

[1]Department of Ophthalmology, Addis Ababa University, Addis Ababa, Ethiopia
[2]Child Health Evaluative Sciences Program and Centre for Global Child Health, SickKids Research Institute, Toronto, Ontario, Canada
[3]Department of Pediatrics and Child Health, Addis Ababa University, Addis Ababa, Oromia, Ethiopia
[4]Paediatrics and Child Health, College of Health Sciences, Addis Ababa University, Addis Ababa, Ethiopia
[5]Ophthalmology and Vision Sciences, The Hospital for Sick Children, Toronto, Ontario, Canada
[6]Department of Ophthalmology and Vision Sciences, University of Toronto, Toronto, Ontario, Canada

**Correspondence to**
Dr Sadik Taju Sherief; goge4000@yahoo.com

## ABSTRACT

**Objective** The aim of this study was to determine the prevalence, causes of ocular disorders and visual impairment among preterm children previously admitted to neonatal intensive care units in Addis Ababa, Ethiopia.

**Methods and analysis** A prospective screening survey was conducted from February to June 2019 at the paediatric eye clinic of Menelik II Hospital. Children who were preterm at birth and who attended the eye clinic were included in the study. Data on demographic and neonatal characteristics, neonatal and maternal comorbidities and ocular disorders were collected. OR and univariate analysis were used to identify predictors of ocular diseases and visual impairment.

**Results** There were 222 children included in the study with a mean age at presentation of 2.62 years (range 2.08–6.38 years), mean gestational age 34.11 weeks (range 30–36) weeks and mean birth weight 1941.72 g (range 953–3500 g). Nearly two-thirds had ocular disorders with refractive error (51.8%), strabismus (11.3%) and a history of retinopathy of prematurity (ROP) (7.2%) being more common. One-fourth of the children had visual impairment, and the prevalence of amblyopia was 40.1%. Uncorrected refractive errors, strabismus and ROP were causes for visual impairment.

**Conclusion** Visual impairment and amblyopia are common in Ethiopia. There is a need to develop a screening protocol for ocular disorders for preterm children to enhance early detection and prevention of childhood visual impairment.

## WHAT IS ALREADY KNOWN ON THIS TOPIC

⇒ In many low-income and middle-income countries, the survival of preterm infants has improved as neonatal systems have improved.
⇒ Preterm children are at a higher risk of developing ocular disorders, visual impairment and amblyopia than term children.

## WHAT THIS STUDY ADDS

⇒ The magnitude and causes of ocular morbidity among preterm children are not well studied in sub-Saharan African countries. This study of preterm children admitted to two neonatal intensive care units in a sub-Saharan country shows that preterm infants develop a higher rate of visual impairment and amblyopia.

## HOW THIS STUDY MIGHT AFFECT RESEARCH, PRACTICE OR POLICY

⇒ The findings of this study provide some evidence for screening for ocular diseases in preterm children, but further studies are needed.
⇒ A follow-up prospective study commencing in 5-year time would be of value as the number of surviving very low birth weight infants may significantly increase.

## INTRODUCTION

Global, regional and national estimates of preterm birth (defined as childbirth at less than 37 completed weeks) using the 2019 Global Burden of Disease study showed 15.22 million preterm births.[1] In the Global Burden of Disease Study, 3.1% of all disability-adjusted life years were attributed to preterm birth, similar to the burden of HIV or malaria.[2] More than 95% of preterm births are occurring in developing countries. Globally the estimated preterm birth rate is 11.1%. Over 60% of preterm births occur in sub-Saharan Africa and South Asia.[1] Ethiopia belongs in the top 15 countries that contribute to two-thirds of

the world's preterm babies with a preterm rate of 14.1% out of 481 deliveries.[3]

From 6 months of pregnancy to term is considered the most active period for ocular development.[4] Improved neonatal care has increased the survival rates of extremely preterm infants with birth weights (BWs) of 1000 g or gestational age (GA) of 28 weeks; at the same time, retinopathy of prematurity (ROP) has become a significant threat to visual function.[5–7]

Preterm children are reported to have an increased incidence of visual impairment because of perinatal lesions in the brain.[8–10]

It has been noted that both preterm birth and ROP have an effect on the developing visual system, leading to decreased visual acuity, decreased contrast sensitivity (CS) and

an increase in colour vision deficiencies.[11–16] Population-based studies suggest that ophthalmic impairments remain common in very low BW infants.[11 16 17] Effects of prematurity on ocular and neurological development include ROP, refractive error, strabismus, cerebral visual impairment, colour vision deficits, reduced CS, visual field defects and decreased visual acuity.[16] According to population studies, the incidence of ROP, whether moderate or severe, for infants born at less than 1500–1700 g ranges from 22% to 49%.[17–19]

In a cohort study, children with lower BWs had significantly worse near and distance visual acuity at ages 10–12 years compared with full-term infants.[10] Additionally, infants born prematurely without ROP are more likely to have myopia and anisometropia than infants born at term because preterm babies are more likely to experience refractive errors.[20]

An increased incidence of strabismus has also been reported in children born prematurely, regardless of the presence of ROP.[21–24]

Research on ocular morbidities among preterm infants in sub-Saharan African nations is limited. Before 2020, blindness from ROP was not reported in Ethiopia, including studies based on schools for the blind.[25 26]

To determine the top causes of illness and mortality in preterm infants admitted to neonatal intensive units (NICUs) in Ethiopia, an Ethiopian Study of Illness in Preterms (SIP) was conducted based on standardised diagnostic protocols. This study is part of the SIP focusing on ocular morbidities among preterms. The present study aimed to identify ocular disorders in a population of preterm children with and without ROP.

## METHODS
### Study design and subjects
The SIP is a prospective study conducted to determine the top causes of illness and mortality in preterm infants admitted to hospitals in Ethiopia based on standardised diagnostic protocols.[27] The study participants of this current study are from the SIP which was conducted from February to June 2019. Patients or the public were not involved in research design, conduct, reporting or dissemination plans.

### Study setting
For the SIP, standard protocols were developed to undertake a physical examination and laboratory investigation, in particular microbiology, radiologic and ultrasound examinations.

There were initial and follow-up examinations to detect the progress of the preterm infant. Addis Ababa University, Gondar University, Jimma University and St. Paul Millennium Medical College were included in the SIP. However, for this ocular morbidity aspect of the SIP, only preterm children from Addis Ababa, Tikur Anbessa Hospital, Gandhi Hospital and St. Paul Millennium Medical College were included.

### Recruitment methods
Inclusion criteria were (1) GA<37 weeks and (2) participation in the SIP. The preterm children were identified from the SIP database. Parents of all preterm infants received a phone call invitation to participate in our investigation.

### Assessment of prenatal and postnatal history
History data were assessed from each child's recorded file for the enrolled children. The following antenatal risk factors were extracted: maternal age, in vitro fertilisation, antenatal corticosteroids, pre-eclampsia/eclampsia, diabetes, HIV/AIDS, chorioamnionitis, mode of delivery and multiple births. Neonatal factors extracted included sex, GA, BW, resuscitation in the delivery room, respiratory distress syndrome, duration of invasive/non-invasive mechanical ventilation and oxygen therapy, intracranial haemorrhage, patent ductus arteriosus, neonatal sepsis, necrotising enterocolitis, number of blood transfusions and bronchopulmonary dysplasia. There were no regular ROP screening programmes within the NICUs of the hospitals where the patients were admitted. There was no referral system from the NICUs to the ophthalmology clinic, except if the parents noted a concern. In addition, all parents were interviewed using a standardised protocol to request information concerning medical history of the child and parents, including ocular and general morbidities.

### Definitions
GA was determined using last menstrual period, Ballard and Dubowitz scores and ultrasound assessment. Studies in Papua New Guinea have shown good concordance (0.878, 0.914 and 0.886, respectively) compared with antenatal ultrasound as the gold standard.[28] LMP in a low-resource setting such as Bangladesh was found to be a more reliable measure of GA than previously thought for the estimation of postnatal GA of preterm infants.[29]

Preterm infants were further classified as late and moderate preterm (32 to <37 weeks), very preterm (28 to <32 weeks) and extremely preterm (less than 28 weeks). Glasses were prescribed if there was myopia>1.0 D, astigmatism>1.0 D or hypermetropia>+2.0 D.

### Eye examination
All examinations were performed by the PI and lead author (STS), a paediatric ophthalmologist. Testing of best-corrected visual acuity was performed with Lea symbols until school enrolment, and after that, ETDRS (Early Treatment Diabetic Retinopathy Study) was used in all subjects. In cases of visual acuity below 6/60, depending on the children's age, Lea symbols or Landolt rings were used at a distance of 1 m. Values were converted for analysis into the logarithm of the minimum angle of resolution (logMAR).[30]

Cyclopentolate (0.5%) eye drops were administered three times at 10 min intervals, after which cycloplegic refraction and keratometry were analysed with

an autorefractor (Nidek ARK-1s keratometer, Japan). The spherical equivalent refractive error was calculated by adding the spherical value and half of the cylindrical value. Anisometropia was defined as a difference between the patients' eyes of ≥1.5 D of spherical equivalent. Orthoptic examination for strabismus included the cover–uncover test and alternate cover test, the Hirschberg Test and examination of fixation behaviour, as well as the presence or absence of nystagmus after having corrected refractive errors. If a child presented with heterotropia, an alternating prism cover test was added to measure the angle of deviation in prism dioptres.

Strabismus was defined as constant or intermittent heterotropia of any dimension at a distance and/or near fixation after correcting refractive error. Classification of strabismus was categorised depending on deviation from the primary position (esotropia or exotropia). An anterior segment examination was done using slit lamp biomicroscopy. A dilated posterior segment examination was conducted using indirect ophthalmoscopy with a 28 D lens. ROP was diagnosed retrospectively from the patients' chart.

Data analysed using IBM SPSS V.21.0 (SPSS). Continuous variables were expressed as the mean±SD or as the median when appropriate. Categorical variables were expressed as proportions. The $\chi^2$ test was used to analyse the association between categorical variables. Associations between ocular morbidities and continuous and categorical variables were computed using Fisher's exact test and Pearson $\chi^2$ test, respectively. Continuous variables were compared using ANOVA. Values of $p < 0.05$ were considered statistically significant.

## RESULTS
During the study period, 222 infants (146 from Saint Paul Hospital and 76 from TASH) were included in this study.

### Characteristics of the study population
Slightly more girls than boys were screened (52.7% and 47.3%, respectively). The majority of the study participants (n=156, 70.3%) were less than 3 years of age and the mean age at presentation was 2.62±0.49 years (range 2.08–6.38). In total, 123 of the 222 children (55.4%) had a GA≤34 weeks and 43 (19.4%) had a BW≤1500 g. BW ranged from 953 to 3500 g with a mean of 1941.72 g (SD 445.49), and GA ranged from 30 to 36 weeks with a mean of 34.11 weeks (SD 1.47). In total, 123 children (55.4%) were delivered vaginally and 80 (36.1%) had multiple gestations. In total, 48 children (21.7%) were born to mothers with pregnancy-induced hypertension, and 8 (3.7%) mothers tested positive for HIV (table 1).

The mean BWs of children from Saint Paul Hospital (SPH) and Tikur Anbessa Specialized Hospital (TASH) NICUs were 1888.5±403.6 (953–3000) g and 2043.94±503.74 (1125–3500) g, respectively; mean GAs were 34.14±1.49[30–36] weeks and 34.08±1.44[30–36] weeks,

respectively. Differences in these parameters were not statistically significant (table 2).

### Ocular morbidities and risk factors
Overall, 145 (65.3%) of the children had ocular disorders at the presentation, of which 92 (63.4%) had isolated ocular diseases (69 refractive error, 13 nasolacrimal duct obstruction, 5 strabismus and 5 ROP). The mean age at presentation of children with ocular disorders was 2.7±0.5 (2.1–6.4) years, and there were more girls with a male-to-female ratio of 1:1.27. None of the eyes examined had anomalies of the anterior segment or lens.

The mean GA was 34.14±1.49[30–36] weeks, and BW was 1927.27 ± 429.19 (953–3100) g. Refractive errors were the leading type of ocular morbidity seen in 115/222 (51.8%), followed by NLDO (Nasolacrimal duct obstruction) (21.2%) (table 2).

### Refractive error
In total, 115 (51.8%) children had a refractive error, of which 55.5% (81/146) and 44.7% (34/76) of children enrolled from the SPH and TASH had refractive errors, respectively.

The mean age at presentation was 2.68±0.56 (2.08–6.38) years, and the male-to-female ratio was 1:1.25. In total, 39 (59%) of children aged >3 years developed refractive error in comparison with 76 (48.1%) of those aged <3 years.

The mean GA and BW of children with refractive errors was 34.11±1.54[30–36] weeks and 1892.34±414.55 (1080–3100) g, respectively. Myopia was the most common type of refractive error, accounting for 78/115 (60.8%) of cases, followed by astigmatism (30, 26.1%) and hyperopia (15, 13.1%). Gender, GA, BW, oxygen supplementation, children and maternal morbidity were not statistically associated with refractive error (table 3).

### Strabismus
In total, 25 children (11.3%) had strabismus (5 isolated and 20 in combination with refractive error, nystagmus, ROP and nasolacrimal duct obstruction). The age at presentation was 2.73±0.52 (2.1–3.6) years, and the male-to-female ratio was 1.08:1.

The mean GA and BW were 34.0±1.41[30–36] weeks and 1906.76±489.92 (1140–3000) g, respectively. In total, 13 children had esotropia, and the rest had exotropia. There was no statistically significant association between GA, BW and strabismus (table 3). In this study, the prevalence of strabismus among children aged >3 years was 16.7% compared with 8.9% in those <3 years. However, older age was not statistically associated with strabismus.

### Retinopathy of prematurity
Previous history of ROP was noted in 16/222 (7.2%) of the children enrolled in this study. Most patients (12, 57%) with ROP were from SPH. Almost all of them (15/16) had a GA<34 weeks, and the mean GA and BW of patients with ROP were 32.19±1.33[30–35] weeks and 1596.25±483.64 (953–2600) g, respectively.

**Table 1** Characteristics of premature children and mothers screened for ocular disorders

| Variable | | Total N | Male N | % | Female N | % |
|---|---|---|---|---|---|---|
| Birth weight | ≤1500 g | 43 | 20 | 46.5% | 23 | 53.5% |
| | >1500 g | 179 | 85 | 47.5% | 94 | 52.5% |
| Gestational age | ≤34 weeks | 122 | 60 | 49.2% | 62 | 50.8% |
| | >34 weeks | 100 | 45 | 45% | 55 | 55% |
| Multiple gestation | Yes | 80 | 42 | 52.5% | 38 | 47.5% |
| | No | 142 | 63 | 44.4% | 79 | 55.6% |
| Oxygen supplementation | Yes | 97 | 47 | 48.5% | 50 | 51.5% |
| | No | 125 | 58 | 46.4% | 67 | 53.6% |
| Infantile morbidity | Sepsis | 6 | 2 | 33.3% | 4 | 66.7% |
| | IVH | 2 | 0 | 0% | 2 | 100% |
| | BPD and sepsis | 1 | 1 | 100% | 0 | 0% |
| | None | 213 | 102 | 47.9% | 111 | 52.1% |
| Mode of delivery | Vaginal delivery | 123 | 53 | 43.1% | 70 | 56.9% |
| | Caesarean section | 99 | 52 | 52.5% | 47 | 47.5% |
| Multiparity | Yes | 47 | 18 | 38.3% | 29 | 61.7% |
| | No | 175 | 63 | 34.3 | 59 | 65.7% |
| Maternal morbidity | PIH | 44 | 22 | 50% | 22 | 50% |
| | HIV | 5 | 2 | 40% | 3 | 60% |
| | HIV and PIH | 3 | 3 | 100% | 0 | 0% |
| | DM | 2 | 0 | 0% | 2 | 100% |
| | DM and PIH | 1 | 0 | 0% | 1 | 100% |
| | TORCH | 1 | 1 | 100% | 0 | 0% |
| | None | 166 | 77 | 46.4% | 89 | 53.6% |
| NICU location | SPH | 146 | 65 | 44.5% | 81 | 55.5% |
| | TASH | 76 | 40 | 52.6% | 36 | 47.4% |

BPD, bronchopulmonary dysplasia; DM, diabetes mellitus; IVH, intraventricular haemorrhage; PIH, pregnancy-induced hypertension; SPH, Saint Paul Hospital; TASH, Tikur Anbessa Specialized Hospital; TORCH, toxoplasmosis, rubella, cytomegalovirus, herpes simplex and HIV.

Nine patients with ROP had an associated refractive error (six myopia and three astigmatism). Only one patient had an associated intermittent exotropia.

In univariate analysis, ROP was statistically associated with low GA and low BW (table 3). Multivariable logistic regression analysis was not conducted due to the small number of children with ROP.

**Table 2** Types of ocular disorders among premature children screened

| Ocular disorders | N | % |
|---|---|---|
| Refractive error | 115 | 51.8 |
| Nasolacrimal duct obstruction | 47 | 21.2 |
| Strabismus | 25 | 11.3 |
| Retinopathy of prematurity | 16 | 7.2 |
| Others | 5 | 2.3 |

Some ocular disorders occurred more than once.

## Visual impairment and ocular disorders

The mean VA of the right and left eyes was 0.22 (SD 0.23) logMAR and 0.17 (SD 0.21) logMAR, respectively. The mean VA in the better and worse eyes was 0.17 (SD 0.22) logMAR and 0.28 (SD 0.21) logMAR, respectively. In this study, 101 (45.9%) and 181 (81.5%) of the children had subnormal visual acuity (>logMAR 0.1) in the better and worst eyes, respectively.

Nearly one-fourth (55, 24.8%) of children screened had visual impairment in the better eye. Of this group, 51 (92.7%) had uncorrected refractive error alone (34/51) or with strabismus (10/51), ROP (6/51) or nystagmus (1/51). In total, 89 (40.1%) patients had amblyopia, of which 59/89 (66.3%) had bilateral amblyopia from uncorrected refractive error. Isometropic and anisometropic amblyopia from uncorrected refractive error were the most common causes of amblyopia, contributing to 49/89 (55%) and 20/89 (22.8%) of cases, respectively. Of the 16 cases with ROP, 12 (75%) had a visual impairment

**Table 3** Ocular disorders by sex, gestational age (GA) and birth weight among premature children screened for ocular disorders

| Type of disorder | Variables | | Yes N | No N | OR | P value |
|---|---|---|---|---|---|---|
| Refractive error | Sex | Male | 51 | 54 | 0.78 (0.46–1.33) | 0.361 |
| | | Female | 64 | 53 | | |
| | BW | ≤1500 g | 22 | 21 | 1.03 (0.53–2.01) | 0.926 |
| | | >1500 g | 93 | 86 | | |
| | GA | ≤34 weeks | 62 | 60 | 1.09 (0.64–1.85) | 0.746 |
| | | >34 weeks | 53 | 47 | | |
| Strabismus | Sex | Male | 12 | 93 | 1.03 (0.4501.37) | 0.940 |
| | | Female | 13 | 104 | | |
| | BW | ≤1500 g | 7 | 36 | 0.57 (0.22–1.48) | 0.246 |
| | | >1500 g | 18 | 161 | | |
| | GA | ≤34 weeks | 14 | 108 | 0.95 (0.41–2.2) | 0.911 |
| | | >34 weeks | 11 | 89 | | |
| ROP | Sex | Male | 6 | 99 | 0.65 (0.22–1.85) | 0.415 |
| | | Female | 10 | 107 | | |
| | BW | ≤1500 g | 7 | 36 | | 0.01 |
| | | >1500 g | 9 | 170 | 0.27 (0.09–0.78) | |
| | GA | ≤34 weeks | 15 | 107 | | 0.001 |
| | | >34 weeks | 1 | 99 | 0.72 (0.09–0.56) | |

BW, birth weight; GA, gestational age; ROP, retinopathy of prematurity.

associated with other disorders like refractive error, strabismus and nystagmus.

In univariate analysis, visual impairment in the better eye was statistically associated with ROP, uncorrected refractive error and strabismus with p values of 0.001, 0.001 and 0.004, respectively. Amblyopia was not statistically associated with low GA or low BW (table 4).

## DISCUSSION

The present prospective study examines the effects of prematurity on visual acuity and ocular disorder in children born preterm. In sub-Saharan Africa, neonatal death has decreased by 40% since 1990 due to improved newborn care, likely leading to an increase in childhood ocular morbidity and blindness from

**Table 4** The presence of visual impairment by types of ocular disorders among premature children screened for ocular disorders

| Variables | | Visual impairment in the better eye | | OR (95% CI) | P value |
|---|---|---|---|---|---|
| | | Yes | No | | |
| Sex | Male | 23 | 82 | 0.75 (0.40 to 1.38) | 0.348 |
| | Female | 32 | 85 | | |
| BW | ≤1500 g | 14 | 29 | 0.62 (0.29 to 1.27) | 0.188 |
| | >1500 g | 41 | 138 | | |
| GA | ≤34 weeks | 28 | 94 | 1.24 (0.67 to 2.29) | 0.294 |
| | >34 weeks | 27 | 73 | | |
| Refractive error | Yes | 48 | 67 | 10.24 (4.37 to 23.97) | 0.001 |
| | No | 7 | 100 | | |
| Strabismus | Yes | 12 | 13 | 3.31 (1.41 to 7.77) | 0.004 |
| | No | 43 | 154 | | |

BW, birth weight; GA, gestational age.

diseases like ROP.[32] Despite this positive progress, data on the extent of ocular diseases among the preterms in sub-Saharan Africa are limited. Our study has demonstrated that the prevalence of ocular diseases and visual impairment in Ethiopian children born preterm is high. To our knowledge, this is the first study to assess the prevalence and causes of ocular disorders and visual impairment among children born preterm and admitted to NICUs.

In Ethiopia, intensive neonatal care has expanded in many public and private hospital NICUs since 2013,[33] and neonatal mortality per thousand live births has declined modestly from 39 in 2000 to 33 in 2019.[34] A prospective screening survey among neonates admitted to two NICUs in Ethiopia showed that 32.2% of the screened infants had any stage ROP.[35] However, there is no regular ROP screening programme in the country. A comparison of studies of ocular morbidity and visual impairment among preterm children is difficult as there are methodological variations such as differing age groups, inclusion or exclusion of ROP, stages of ROP and cohort size.

Even though genetic and visual experiences predominantly determine the prevalence of refractive error, studies have shown that low BW interrupts emmetropisation and increases the prevalence of refractive error.[36] In our study, nearly half of the premature screened children (51.8%) had refractive error, which is comparable to a survey from Turkey (53.8%)[37] but higher than in Italy (42.3%)[38] and in cohorts of extremely preterm infants from Sweden (29.7%)[39] and Norway (10%).[40] In our study, the prevalence of myopia was 35.1%, which was higher than a cohort of preterm children at age 10–12 years from the UK (18.9%),[41] India (15.8%)[42] and Sweden (4.1%).[39] The prevalence of hyperopia in our study, 13.5%, is comparable with that reported in Turkey (14.3%)[37] and Sweden (17.1%)[39] but higher than the UK (6.6%)[41] and India (8.54%).[42] In the present study, 13.5% of the preterm children had clinically significant astigmatism, which was lower than that reported in Norway (21%)[40] and India (55.6%)[42] and higher than in Turkey (5.7%)[37] and Sweden (6.5%).[39] The higher proportion of myopia seen in our study, in comparison with studies from the UK,[41] India[42] and Sweden,[39] is supported by long-term studies which have confirmed the increased incidence of myopia following preterm birth.[43]

Manifest strabismus was seen in 11.3% of our cohort, which is comparable to studies from Norway (10%),[40] the UK (13.6%),[15] Sweden (13.5%)[20] and Australia (14%),[44] and lower than reported in Sweden (17%),[36] the UK and Ireland (24%)[45] and Germany (26%).[46] It is unclear at what age the different types of strabismus develop,[36] and the age at onset of strabismus in low BW children is variable, from the first few months of life to many years later.[11 15 16 21 22 47] In our study, a higher prevalence of strabismus in those aged >3 years was noted. This finding (16.7%) is comparable with a similar age group from Sweden.[20] Regarding the type of strabismus, we detected similar proportions for esotropia and exotropia.

This is similar to the other studies from Germany[46] and England.[41] However, other investigations confirmed that esotropia was the most frequent type of strabismus[20 39 48] The increased prevalence of strabismus in the low BW population is well documented (21, 36 and 44). Such an association was not apparent in our study, as most of the children were considerably higher in weight and older than in the studies mentioned above.

The prevalence of ROP in our study is 7.2%, lower than in other studies, from sub-Saharan African countries, including Ethiopia, which ranged from 15% to 41.7%.[35 49–51] The lower prevalence of ROP in our study can be explained by our data collection method, where we depend on the history of ROP either from the patient's parents or from old features of ROP.

In the present study, 46% of the children had subnormal visual acuity (>logMAR 0.1) in the better eye, which is comparable with a population-based study from Norway (45.9%).[40] The figure is higher than what has been reported for prematurely born children with BWs 1500–2000 g (15 %) from Denmark[9] and from Sweden 32%.[20] Birch *et al* reported significantly lower visual acuities in low BW infants compared with those born full term.[52] In our study, there was no statistically significant correlation between BCVA (Best Corrected Visual Acuity) and BW or GA, similar to a study from Turkey.[37] However, Dowdeswell *et al*[53] found low levels of distance visual acuity in preterm children compared with full-term children. However, in our study, ocular morbidities like strabismus, refractive error and ROP were statistically associated with visual impairment.

In our study, the prevalence of amblyopia among premature children was 40.1%. The result in our study is much higher than other studies from Australia (7.3%)[44] and Turkey (7.7%).[37] Previous studies have shown that prematurity and low BW are two risk factors for amblyopia.[41 54] Nevertheless, amblyopia was not statistically associated with low GA and BW. Even if we did not find a statistical association between GA and BW with amblyopia, the prevalence among premature children is higher than in other studies; this indicates that more importance should be given to screening amblyopia risk factors for premature infants.

The strengths of this study were the prospective controlled study design with a high number of participants, the multicentre design which increases the representativeness of our research and the availability of medical information from all children and mothers, which allowed a very detailed examination and an adjustment for different possible confounding factors. The strict standardisation reduced the probability of examiner-dependent variances.

Limitations of the study included the wide age range of the examined children, some of whom were at an early age and phase of refractive development, and other older children that can affect the physiologic refractive changes noted in normal health children. The other limitation is there is a chance that those infants with poor health

outcomes did not take part in our study. In subsequent research, we will continue following up with these infants to determine future changes in their refractive error and strabismus.

## CONCLUSION

In conclusion, the rates of ocular disorders, visual impairment and amblyopia in these NICUs in Ethiopia were higher than in other studies. Refractive error, strabismus and ROP were all significant risk factors for visual impairment. These findings underline the importance of early screening of premature infants for vision and amblyopia. As the two NICUs included in the survey are Ethiopia's main neonatal referral centres, it can be postulated that ocular morbidities, visual impairment and amblyopia are emerging as potentially avoidable causes of childhood blindness among preterm children in Ethiopia. Developing preterm ocular-related screening protocols within the NICUs, strengthening the referral links between the NICUs and eye centres and further detailed comparative studies between preterm and term children for ocular disorders are recommended.

**Acknowledgements** The authors wish to acknowledge the assistance of the staff of the SIP project (Ahmed, Beleyu, Efrata and Wagaye) for their support during data collection. Our special appreciation goes to Dr Martha H/Mariam and Dr Medhanit Tesfaye from Menelik II Hospital's Pediatrics Ophthalmology Clinic.

**Contributors** Drafting of the manuscript and conception and design of study: STS, LMM, AM and AD. Revision of the manuscript for important intellectual content, data acquisition, analysis or interpretation, approval of final manuscript to be published and read and approved the final version of the manuscript: all authors.

**Funding** The SIP was supported by the Bill & Melinda Gates Foundation (OPP1136965).

**Competing interests** None declared.

**Patient and public involvement** Patients and/or the public were not involved in the design, or conduct, or reporting, or dissemination plans of this research.

**Patient consent for publication** Consent obtained from parent(s)/guardian(s).

**Ethics approval** This study involves human participants and was approved by the ethics committee of Addis Ababa University Ethics Review Committee (Ref No. 003/2016) in line with the relevant national and institutional guidelines on care and clinical research. The research was performed in accordance with the Declaration of Helsinki. All parents or legal guardians provided informed consent before the examination. Participants gave informed consent to participate in the study before taking part.

**Provenance and peer review** Not commissioned; externally peer reviewed.

**Data availability statement** All data relevant to the study are included in the article or uploaded as supplementary information.

**ORCID iD**
Sadik Taju Sherief http://orcid.org/0000-0003-4614-4563

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
