## [Reviewer comments · BMJ Paediatrics Open]

ARTICLE DETAILS

TITLE (PROVISIONAL)	Prevalence and causes of ocular disorders and visual impairment among preterm children in Ethiopia
AUTHORS	SHERIEF, SADIK TAJU Muhe, Lulu M Mekasha, Amha Demtse, Asrat Ali, Asim

VERSION 1 – REVIEW

REVIEWER	Dr. Dupe Ademola-Popoola University of Ilorin Teaching Hospital, Ophthalmology
REVIEW RETURNED	21-Nov-2023

GENERAL COMMENTS	Kindly look at each of the comments in the main manuscript
--

REVIEWER	Frank Martin University of Sydney - Sydney Medical School Nepean, Pediatrics Child Health & Ophthalmology
REVIEW RETURNED	15-Dec-2023

GENERAL COMMENTS	This is a very valuable study. The data reflects what I would have expected from this population. A follow up prospective study commencing in 5 years time would be of value as I expect the number of surviving very low weight infants (500-750 g) to significantly increase. The results from this study would be directly comparable to studies from the developed world. This study provides excellent data to use to advocate for a screening program for all low birth babies
--

VERSION 1 – AUTHOR RESPONSE

Reviewer: 1

Dr. Dupe Ademola-Popoola, University of Ilorin Teaching Hospital

Comments to the Author

Kindly look at each of the comments in the main manuscript

Excuse me; we didn't get the comments.

Reviewer: 2

Frank Martin, University of Sydney - Sydney Medical School Nepean

Comments to the Author

This is a very valuable study. The data reflects what I would have expected from this population. A follow up prospective study commencing in 5 years time would be of value as I expect the number of surviving very low weight infants (500-750 g) to significantly increase. The results from this study would be directly comparable to studies from the developed world.

This study provides excellent data to use to advocate for a screening program for all low birth babies

Thank you so much for the kind words and positive feedback.

Thank you for the opportunity to be considered for publication in your journal. We look forward to your response.

Sincerely,

Dr. Sadik T. Sherief.